# Atelocollagen Application in Human Periodontal Tissue Treatment—A Pilot Study

**DOI:** 10.3390/life10070114

**Published:** 2020-07-16

**Authors:** Marzena Wyganowska-Swiatkowska, Anna Duda-Sobczak, Andrea Corbo, Teresa Matthews-Brzozowska

**Affiliations:** 1Department of Dental Surgery and Periodontology, Poznan University of Medical Sciences, Bukowska 70, 60-812 Poznan, Poland; 2Department of Internal Medicine and Diabetology, Poznan University of Medical Sciences, Mickiewicza 2, 60-830 Poznan, Poland; a_sobczak@onet.pl; 3Private Practice Medical Spa, Via Cassia 1840, 00123 Rome, Italy; info@andreacorbo.it; 4The Chair and Clinic of Maxillofacial Orthopaedics and Orthodontics, Poznan University of Medical Sciences, Bukowska 70, 60-812 Poznan, Poland; tmatbrzo@gmail.com

**Keywords:** skin, gingiva, collagen, connective tissue stimulation

## Abstract

Background: The aim of this study is the clinical observation of gingival tissue condition after atelocollagen injection. Methods: In 18 patients, 97 gingival class I Miller recessions were divided according to recession height, gingival papillae loss and thickness of gingivae. Atelocollagen (Linerase, 100 mg) was injected into keratinized gingivae twice or thrice, at two-week intervals. Results: Statistically significant changes in gingival recession, amount of gingival papillae loss and thickness of gingiva were observed, after both two and three collagen injections. Although the degree (height) of recession decreased and gingival tissue thickness increased with every injection; there was no difference in gingival papillae loss between second and third collagen injections. Conclusions: The injectable form of atelocollagen is a promising material for gingival soft tissue regeneration and stimulation and allows for reduction in the number of procedures and support in a variety of surgical scenarios. This is a pilot study that clinically measures the impact of injected atelocollagen on periodontal tissue biotype, including the thickness of gingivae and gingival papillae regeneration.

## 1. Introduction

Due to the significant deficiency of collagen found in aging skin, the use of collagen as a biostimulant has, therefore, been highly desirable. The interest in collagen increased greatly after the ability to synthetize its human form. In dentistry, the loss of tissues supporting teeth is a major problem, with a considerable impact on quality of life and aesthetic appearance. Gingival recessions, as the result of dental plaque accumulation, aggressive tooth brushing, periodontal diseases and orthodontic treatment, are large-scale for the population’s problem [1].

Gingival connective tissue consists predominantly of fibroblasts (5%), as well as other cells, such as mast cells, macrophages, neutrophilic granulocytes, lymphocytes and plasma cells, present in the lamina propria of oral mucosa. The extracellular matrix contains glycosaminoglicans, such as hyaluronic acid, dermatan sulfate and heparan sulfate, proteoglycans, glycoproteins and fibers, such as reticulin fibers, collagen fibers, oxytalan and elastic fibers. Collagen fibers (65% of volume) play a key role in the gingival architecture of healthy gingiva, as well as in periodontal disease progression [2]. The level of collagen loss is highlighted as the main marker of this process [3].

The early signs of collagen loss in the perivascular area are observed in the first stage of gingivitis (initial lesion). This is due to the fact that almost 70% of collagen is destroyed around the cellular infiltrate at that stage. The main fiber groups affected during this process are circular and dentinogingival fibers. In the established lesion, there is an adverse relation between number of inflammatory cells and the amount of intact collagen [1]. The process starts apicaly to the junctional epithelium and spreads into the periodontal ligament. The activities of various enzymes, such as collagenase and the phagocytosis, are mechanisms associated with collagen degradation.

Collagen is an essential protein of the connective tissue matrix, which influences the migration of keratinocytes to damaged sites of epidermis or epithelium, and it is an important component of accelerating wound healing and tissue regeneration. Over the embryonic period, the predominant type of collagen in gingiva, called the “fetal collagen”, is collagen type III. After birth, collagen type III is gradually replaced by collagen type I, which becomes the dominant type. However, type III is detected in the reticulum, type V in the pericell area and type IV in blood vessel basement membranes [4]. Type I collagen fibers are situated in the deep layer of gingival connective tissues and type III in more superficial layers. There are two α1 and one α2 chains, but three identical α1 chains are the difference in the structure of the triple helix of collagens type I and III. Both collagenous and noncollagenous molecules are combined to create fibrils [5]. Collagen for biomaterial production may be extracted from animals and humans (cadavers, placenta, amniotic sac) [6] or may be synthetized using recombinant techniques [7]. For extraction from animal tissue, two different methods are used: pepsin digestion or acid solubilization, obtaining two different collagen forms: atelocollagen and tropocollagen. Atelocollagen is preferred in commercial use due to the associated cross-species antigenicity of the p-determinant located in the telopeptides [8]. For this reason, both allogenic and xenogenic collagen are widely recognized as safe biomaterials.

Collagen is seen as one of the most valuable biomaterials, due to its excellent biocompatibility, weak antigenicity and its biodegradability. Collagen products may have various forms, such as liquid, gel, membrane and granules. This makes it very useful in medical applications, especially for tissue engineering: oral mucosa, gingival and bone tissue, skin regeneration and stimulation, as well as a scaffold for artificial blood vessels and valves. Currently, injectable collagen is available, not only as a tissue filler, but also as a connective tissue biostimulator. This form of collagen is traditionally used for skin applications; however, similar results may be expected after collagen stimulation in oral mucosa, as it also contains connective tissue.

The clinical observation of the condition of gingival tissue after atelocollagen injection is the aim of this study.

## 2. Methods

### 2.1. Study Design and Population

The study was carried out, in cooperation with the Postgraduate Study in Aesthetic Medicine Poznan University of Medical Sciences, with in-patients referred to the outpatient clinic of the Department of Maxillofacial Orthopedics and Orthodontics, between 2016 and 2017. The study design was approved by the local ethics committee (UMP 919/16) and was conducted according to the guidelines of the Declaration of Helsinki on biomedical research involving human subjects. All participants provided written informed consent before enrolment in the study. Out of all the patients with gingival recessions, only those with class I Miller gingival recessions (the gingival defect not extending to the muco-gingival junction) were included into the study. Exclusion criteria were local inflammation, active periodontitis, systemic diseases and non-caries lesions. In 18 patients (2 male and 16 female) aged 40–55 years, 97 gingival recessions class I Miller were examined. Three outcomes were measured: recession height, gingival papillae loss and thickness of gingiva. The recession height (distance from cemento-enamel junction to gingival margin) and gingival papillae loss (distance from contact point to top of the papilla) were measured with a standardized periodontal probe (UNC-15 probe). For every patient, individual injections and measuring patterns were done. The following measurements were always done before the next injection and during one visit: thickness of gingiva was determined during collagen injection (before first injection and every consecutive one) as the distance between the end of the needle (size 0.4 mm) and the point indicated by a stopper placed on the needle, which was measured with a digital electronic digital caliper (Topex). Point of measurement was defined according to the cemento-enamel junction. In all cases, the atelocollagen (Linerase, 100 mg), thinned with 4.5 mL 0.9% NaCl and 0.5 mL 2% Lignocain, was injected into keratinized gingiva, two millimeters above gingival papillae basement, three times in two-week intervals. The distance between points of injections was above 10 mm.

### 2.2. Statistical Analysis

All statistics were performed with the commercially available software STATISTICA V12.5 PL (StatSoft, Tulsa, OK, USA). The Shapiro–Wilk test was applied to assess distribution of continuous variables. Distributions of all the continuous variables were not normal. Data are presented as median (interquartile range (IQR)) for continuous variables. The differences in gingival indices before and after collagen administration were examined using nonparametric Wilcoxon signed-rank test. *p* < 0.05 was considered statistically significant.

## 3. Results

Among the examined gingival recessions in all 97 cases, the injection of atelocollagen was done twice. In 37 cases of recessions, thrice. Statistically significant changes in gingival recessions, amount of gingival papillae loss and thickness of gingiva were observed after two (Table 1), as well as after three (Table 2), collagen injections. Although the recession height decreased and gingival tissue thickness increased with every injection, there was no difference in the gingival papillae loss between the second and third collagen injections (Table 3). The changes were visible in clinical examination, and thus were clinically significant (Figure 1 and Figure 2).

## 4. Discussion

In general medicine, collagen has advanced uses in drug, protein and nanoparticles delivery system. Moreover, it is a basic matrix for cell culture. It is also considered to be an excellent biostimulant and filler [9]. The main applications are collagen shields and sponges for better healing [10] because of its high compatibility and frame form for young cells of surrounding tissues. Collagen fulfils another parameter (balance between stability and degradability) for clinically used biomaterials.

Collagen in biomaterials degrades faster than natural ones and protects natural collagen from matrix metalloproteinases. The degradation products also stimulate fibroblast migration and proliferation, epithelial and vascular endothelial cells, encouraging granulation tissue formation [11]. It is well known that type I collagen scaffolds are excellent matrix substrates providing adhesive properties for cell proliferation, migration and differentiation. Collagen sponges are used in dental surgery due to their inherent advantages like hemostatic property biocompatibility. They are physiologically metabolized and can be fully resorbed by the host system without any side effects. Moreover, they have a positive effect on blood coagulation and wound healing, and facilitate its maturation and stability by enhancing initial clot and fibrin linkage formation [12]. Other forms of collagen used in regenerative procedures in periodontium are collagen membranes or 3D collagen matrices, such as Mucoderm. The same clinical results were observed when collagen membrane was used as a graft in gingival recession treatment, compared to subepithelial connective tissue graft (CTG) [13]. However according to meta-analysis, there was no evident difference in the width of keratinized tissue, the key parameter for gingival tissue stability, and only a significant gain in the gingival thickness for CTG [14]. Pre-clinical histological studies indicated better results (soft tissue thickness, amount of keratinized gingiva) when collagens, such as collagen matrices (CM), human skin equivalents (bilayered cell therapy (BCT)), human fibroblast-derived dermal substitute (HF-DDS) and acellular dermal matrix graft (ADMG), were used in comparison to a control group [15]. In our study, the injectable form of collagen was used with promising results. Even after two collagen injections, significant improvement of gingiva condition was detected. Both the recession height as well as the papillae improved. The thickness of gingivae significantly increased. The comparison and assessment of our results is however impossible, due to the current absence of similar studies. The collagen used in our study is produced for aesthetic medicine and characterized as equine atelocollagen.

The connective tissue of skin varies with masticatory mucosa, which covers the hard palate and gingiva by loose organization and has no attachment to the underlying bone [16,17]. This biological stability may reduce wound contraction and faster cell migration [18]. It results in accelerated wound healing in the oral mucosa. Moreover, it is interesting that a distinct phenotype of gingival fibroblasts may have an impact on gingival wound healing [19].

The fibroblasts heterogeneity is visible when comparing the oral mucosa to attached and free gingiva, as well as within a sample taken from one place. Fibroblasts are one of the most copious stromal cell types and normally are the quiescent cell, but after tissue injury, they become activated by the secretion of soluble cell mediators and degraded extracellular matrix [20]. The activated fibroblasts are responsible for tissue repairing by producing and remodeling extracellular matrix components and also stimulation of granulation tissue formation and angiogenesis [21]. Fibroblasts are very perceptive, mostly to changes in the surrounding matrix, growth factors and cytokines and respond to these signals. Moreover, the combination of atelocollagen and fibroblast growth factor is more effective in treatment for skin resistant wounds with deep cavities [22].

This fibroblasts reactivity is key during stimulation by collagen biomaterials. Especially important are the proportions of different types of collagens in injected biomaterials for the size of new collagen fibrils synthesis. The differences involve collagen types I, III and V. The other observation suggests that on the veering materials the cell attachment occurs at different speeds. It is suggested that cells were absorbed deeper into the porcine and equine scaffolds due to different wettability and pore size. Moreover, bovine and equine collagen show the highest cell proliferation [23].

Recently, the equine collagen, because of its more loosely organized structure than porcine collagen, seems to be the safest. This structure enables the lowering of the protolithic enzymes and, consequently, the immunogenicity. The levels of different amino acids within the collagen used for procedures are similar. Levels of proline and hydroxyproline in chicken feet, bovine, porcine skin and bird feet collagen ranges from 17 to 19% [24]. Marine collagen types, because of their lower content of hydroxyproline and, consequently, lower denaturation temperature (25–30 °C), differentiate from mammalian collagen types [25]. Because of its low thermal stability, fish collagen was used in tissue engineering with limited success. Recently, a new collagen type I with higher temperature (37 °C) was purified from tilapia fish scales [24]. It is suggested that this type of collagen is even better than porcine skin collagen I as it stimulates human mesenchymal stem cells [25]. More importantly, the new collagen type is safe during intracutaneous and topical application [26]. The other study indicated that the porcine atelocollagen is safe and has the same effectiveness as bovine atelocollagen, however it does not need skin testing before application [27].

Atelocollagen is one of the collagen types achieved in the collagen extraction process grounded on digestion with pepsin, which allows reducing antigenic epitopes contained in telopeptides, responsible for immunogenicity. Atelocollagen was used for the first time in regenerative medicine already in the 1970s. Its rare properties, such as low immunogenicity, liquid state at 4 °C, solid state at 37 °C and strong positive charge (pI 9), allowed atelocollagen to be used as a carrier of negatively charged proteins and nucleic acids and expanded the field of atelocollagen application on dentistry and regenerative medicine. The porous hydroxyapatite/collagen scaffold consisted of hydroxyapatite nanocrystals and type I atelocollagen as a carrier for bone morphogenetic protein (BMP) in bone regeneration, as well as a carrier for fibroblast growth factor-2 (FGF-2) in osteochondral regeneration [28]. Atelocollagen in connection with silicon membrane and FGF are effective in tympanic membrane regeneration [29,30]. However, type I atelocollagen alone maintains a chondrocyte phenotype and as a gel represents a good carrier for cultured chondrocyte transplantation [31,32]. It also permits proliferation, matrix synthesis and differentiation of mesenchymal stem cells [33]. In connection with mitomycin C, an antitumor agent, it inhibits the proliferation of various cells, and those properties allow controlling scar tissue formation [34]. Atelocollagen sponge was described as promoting fibroblast aggregation, proliferation and formation of collagen fiber bundles in cleft palate closure [35]. Atelocollagen-mediated systemic delivery of siRNA without chemical modifications did not cause any immunostimulation [36] in both animals and human peripheral blood mononuclear cells (PBMCs) [37].

Interestingly, in cosmetic makeup products, the atelocollagen is frequently used in monomeric form, while for cell scaffolds, its polymeric form is preferred [8].

This study has several limitations. Our study was conducted on the small group of patients and only when the recession was no greater than 1 mm. The observation time was 10 weeks and would therefore benefit from a longer time scale. Nevertheless, this is the first clinical examination of gingival tissue outcome done after gingival recession treatment with the injectable form of atelocollagen. The presented results are part of the clinical observation of the impact of equine collagen on periodontal tissue.

## 5. Conclusions

Collagen is regarded as being beneficial in blood coagulation, as the collagen fibrils enable platelets to adhere on the surface and in this way enhance the natural aggregation and degranulation, and initiate the natural platelet plug formation. Through the relationship between hyaluronic acid, enamel matrix proteins and intercellular communications, collagen plays the crucial role in regulating skin and mucosa physiology. It is a safe and biocompatible connective tissue stimulator, which, in injectable form, may have an influence on the gingiva condition through changes in fibroblast proliferative potential. The injectable form of atelocollagen is a promising material for gingival soft tissue regeneration and stimulation and allows for reduction in the number of procedures and support in a variety of surgical scenarios. This is the first clinical examination of gingival tissue outcome done after gingival recession treatment with the injectable form of atelocollagen. All the cases were examined by one periodontologist according to the same protocol. The limitation of this study is the smaller group of patients (37) with triple dosage in relation to the group of patients (97) with double dosage.

## Figures and Tables

**Figure 1 life-10-00114-f001:**
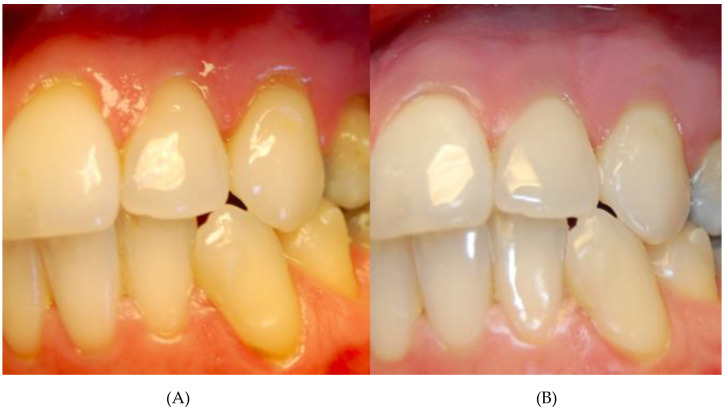
Patient with multiple class I Miller recessions in upper before (**A**) and after thrice atelocollagen injection (**B**).

**Figure 2 life-10-00114-f002:**
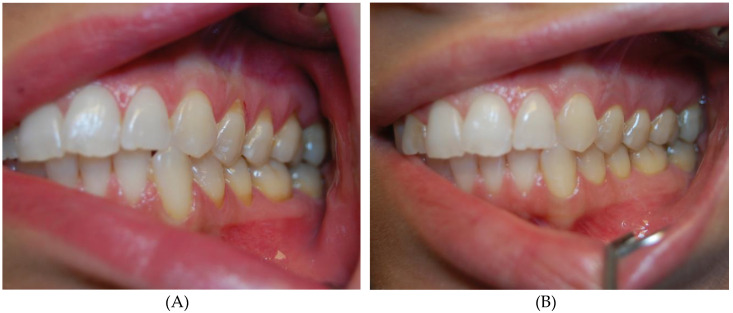
Patient with class I Miller recessions in upper and lower arch before (**A**) and after thrice atelocollagen injection (**B**).

**Table 1 life-10-00114-t001:** Comparison of gingival outcomes before and after two administrations of collagen. Median (interquartile range (IQR)). Wilcoxon signed-rank test. n = 97.

	Before Collagen Injection	After Two Collagen Injections	*p*
Recession high	1 (0.5–2)	0 (0–0.5)	<0.000001
Papillae loss	1 (0.5–1.5)	0 (0–0.5)	<0.000001
Thickness of gingiva	0.3 (0.1–0.5)	0.6 (0.3–0.8)	<0.000001

**Table 2 life-10-00114-t002:** Comparison of gingival outcomes before and after three administrations of collagen. Median (IQR). Wilcoxon signed-rank test. n = 37.

	Before Collagen Injection	After Three Collagen Injections	*p*
Recession high	0.5 (0–1.5)	0 (0)	0.00002
Papillae loss	1 (0.5–1.5)	0 (0)	0.00001
Thickness of gingiva	0.1 (0.1–0.4)	0.7 (0.5–1)	0.000002

**Table 3 life-10-00114-t003:** Comparison of gingival outcomes after two and three administrations of collagen. Median (IQR). Wilcoxon signed-rank test. n = 37.

	After Two Collagen Injections	After Three Collagen Injections	*p*
Recession high	0 (0–0.5)	0 (0)	0.02
Papillae loss	0 (0–0.5)	0 (0)	0.1
Thickness of gingiva	0.5 (0.3–0.6)	0.7 (0.5–1)	0.000004

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
