# Peer review of "Atelocollagen Application in Human Periodontal Tissue Treatment—A Pilot Study"

_life, 2020, doi:10.3390/life10070114_

Round 1

Reviewer 1 Report

The Authors evaluated the gingival tissue conditions after applications of atelocollagen into keratinized gingiva injection.

The manuscript addresses an interesting topic.

Minor issues:

The study designs for experimental and analytic observational studies have the following components:

  • A defined population (P) from which groups of subjects are studied
  • Outcomes (O) that are measured
  • Interventions (I) or exposures (E) that are applied to different groups of subjects

The Authors should better specified and clarified the study design of this paper.

Reviewer 2 Report

This manuscript entitled "Atelocollagen application in human periodontal
tissue treatment – a pilot study" is a nice approach that could be useful in dental surgery. It could be beneficial tool for future dental treatments.
